# Risk Factors for COVID-19 Morbidity and Mortality in Institutionalised Elderly People

**DOI:** 10.3390/ijerph181910221

**Published:** 2021-09-28

**Authors:** Ander Burgaña Agoües, Marta Serra Gallego, Raquel Hernández Resa, Beatriz Joven Llorente, Maria Lloret Arabi, Jessica Ortiz Rodriguez, Helena Puig Acebal, Mireia Campos Hernández, Itziar Caballero Ayala, Pedro Pavón Calero, Montserrat Losilla Calle, Rosario Bueno Nieto, Laura Oliver Messeguer, Rosa Madridejos Mora, Rosa Abellana Sangrà, Tomás M. Perez-Porcuna

**Affiliations:** 1Atenció Primària, Fundació Assitencial Mútua Terrassa, 08221 Terrassa, Spain; aburgana@mutuaterrassa.cat (A.B.A.); martaserra@mutuaterrassa.es (M.S.G.); rhernandez@mutuaterrassa.es (R.H.R.); bjoven@mutuaterrassa.es (B.J.L.); mlloret@mutuaterrassa.cat (M.L.A.); jortiz@mutuaterrassa.cat (J.O.R.); hpuig@mutuaterrassa.es (H.P.A.); mcampos@mutuaterrassa.es (M.C.H.); icaballero@mutuaterrassa.cat (I.C.A.); ppavon@mutuaterrassa.cat (P.P.C.); mlosilla@mutuaterrassa.cat (M.L.C.); rosariobueno@mutuaterrassa.es (R.B.N.); loliver@mutuaterrassa.cat (L.O.M.); rmadridejos@mutuaterrassa.es (R.M.M.); tomas.perez.porcuna@gmail.com (T.M.P.-P.); 2Departament Fonaments Clínics, Universitat de Barcelona, 08036 Barcelona, Spain

**Keywords:** frailty, frail elderly, comorbidity, COVID-19, nursing homes, long-term care

## Abstract

Background: SARS-CoV-2 has caused a high mortality in institutionalised individuals. There are very few studies on the involvement and the real impact of COVID-19 in nursing homes. This study analysed factors related to morbidity and mortality of COVID-19 in institutionalised elderly people. Methods: This cohort study included 842 individuals from 12 nursing homes in Sant Cugat del Vallès (Spain) from 15 March to 15 May 2020. We evaluated individual factors (demographic, dependence, clinical, and therapeutic) and those related to the nursing homes (size and staff) associated with infection and mortality by SARS-CoV-2. Infection was diagnosed by molecular biology test. Results: Of the 842 residents included in the analysis, 784 underwent a Polymerase Chain Reaction (PCR) test; 74.2% were women, the mean age was 87.1 years, and 11.1% died. The PCR test was positive in 44%. A total of 33.4% of the residents presented symptoms compatible with COVID-19 and of these, 80.9% were PCR-positive for SARS-CoV-2. Infection by SARS-CoV-2 among residents was associated with the rate of staff infected in the homes. Mortality by SARS-CoV-2 was related to male sex and a greater grade of dependence measured with the Barthel index. Conclusions: SARS-Cov-2 infection in institutionalised people is associated with the infection rate in nursing home workers and mortality by SARS-Cov-2 with sex and greater dependency according to the Barthel index. Adequate management of nursing home staff and special attention to measures of infection control, especially of individuals with greater dependence, are keys for successful management of future pandemic situations.

## 1. Introduction

At the end of 2019, the city of Wuhan, China reported an increase in patients with respiratory infection by a new coronavirus now identified as COVID-19. On 30 January 2020 the World Health Organization declared that the outbreak of COVID-19 constituted an international public health emergency [1] after which the spread of the pandemic led to the registration of more than 173 million cases in 223 countries and more than 3.7 millions deaths by 8 June 2021 [2].

COVID-19 is an acute respiratory disease caused by the new human coronavirus SARS-CoV-2, producing a high mortality in persons over the age of 60 years and in individuals with previous diseases such as cardiovascular disease, chronic respiratory diseases, diabetes, or cancer [3,4]. Long-term health care facilities such as nursing homes and rehabilitation centres provide care to persons often of elderly age and with mental or physical impairment. The residents of long-term care homes represent a vulnerable population with a greater risk of adverse events and infections due to age, comorbidities, and living in close proximity to other persons [5]. Therefore, different international organisations recommended that long-term care facilities should adopt special measures to protect their residents, staff, and visitors [6].

In most countries, older people have been the most severely affected by COVID-19 [7], and those with comorbidities are at particular risk of having severe infection [8] and are at higher risk of dying as a result of the disease [9,10,11]. In one of the largest case series published so far, of 72,314 cases reported by the Chinese Center for Disease Control and Prevention, case fatality was 8.0% in patients 70–79 years old and 14.8% in patients aged ≥80 years. Older adults appear to be more susceptible to the virus with 75% of known infections being in persons aged 50 and over. Hypertension followed by diabetes and cardiovascular diseases were the most common comorbidities seen in COVID-19 positive patients across major epicentres worldwide [12]. Overall, the risk of severe COVID-19 disease or death is higher in older patients and those with previous medical conditions including chronic obstructive pulmonary disease and cardiovascular disease as some of the most relevant predictors [13]. Males, age greater than 65, and smokers may present a greater risk of developing critical or mortal conditions, and comorbidities such as hypertension, diabetes, and cardiovascular disease, and respiratory diseases can also greatly affect the prognosis of the COVID-19 [14]. In addition, individuals over 70 make up 37% of all the known infections by COVID-19, 48% of all hospitalisations, 33% of the patients in intensive care units, and 86% of deaths [15].

In general, nursing homes are semi-closed settings in which residents, family members, and staff interact. One of the first measures taken to mitigate the possibility of infection by SARS-CoV-2 in our nursing home setting was the restriction of family visits with the subsequent psychological consequences [16]. In this situation, the staff became the main entry point for the transmission of this viral infection to the residents. The correlation between the rate of infection and infected cases of the residents agrees with the results of studies analysing the epidemic curves of outbreaks of SARS-CoV-2 infection in nursing homes being related to transmission to residents by staff [17].

In Europe, Spain is one of the countries with the highest percentage of persons over the age of 80 years (6.02% in January 2020) [18]. During the first wave of the pandemic more than 25,000 deaths were registered in nursing homes, representing 66% of the total number of deaths by COVID-19 [19], mainly in older persons and with a mortality rate of 22.1% [20]. These data came to public attention and the authorities adopted exceptional measures in an attempt to protect the population residing in nursing homes [21,22]. By 10 July 2020 a total of 253,908 confirmed cases of COVID-19 and 28,403 deaths had been reported in Spain [23]. In our country, the COVID-19@Spain Study Group evaluated the predictive factors of mortality by COVID-19 in patients admitted to hospital and reported an associated mortality of 54.9% in subjects ≥80 years of age, and in patients with ≥3 comorbidities the mortality was 47.7% [24]. The real data on mortality by COVID-19 in nursing homes of the elderly were initially difficult to obtain, since in some communities only cases confirmed by polymerase chain reaction (PCR) were quantified (34.1% of the total deaths by COVID-19 by 23 June 2020), and during the first months confirmatory PCRs were only performed in hospitalised cases. Indeed, reports carried out in this respect estimate that the mortality in nursing homes was actually 45–68.1% of the total deaths [19,25].

There are very few high-quality studies on the involvement and the real impact of COVID-19 in nursing homes [26]. Scientific societies have elaborated multiple recommendations regarding the management of the pandemic in nursing homes [27,28,29,30,31]. The scarce number of studies on this subject include reports of series of cases focused on the lethality of the virus, the need for early diagnosis of suspicious cases, and early implementation of preventive measures to avoid the introduction and spread of the virus in nursing homes [32], as well as the role of asymptomatic and presymptomatic residents [33,34,35].

With respect to tools to evaluate comorbidities in older persons and their predictive capacity of mortality, in Spain one study concluded that the PROFUND and Charlson indexes are better predictors of risk of death by COVID-19 in nursing homes than CURB-65 [36]. Another study on the risk factors of mortality by COVID-19 in elderly institutionalised persons in Andorra, concluded that male sex, a low Barthel index, and lymphocytopenia were independent risk factors of death by COVID-19 [37].

A Spanish study on the characteristics of nursing homes found that the model of nursing homes best prepared to manage the COVID-19 was that of public nursing homes with less than 25 places [38].

The aim of this study was to analyse the factors related to morbidity and mortality by SARS-CoV-2 infection in persons living in nursing homes from March to June 2020 (first wave) in the municipality of Sant Cugat del Vallès, Barcelona, Spain.

## 2. Materials and Methods

### 2.1. Study Population

The study population included all the residents (842 persons) of the 12 nursing homes for the elderly in the municipality of Sant Cugat del Vallès, a city near Barcelona, in the autonomous region of Catalonia (Spain). The population is about 90,000 inhabitants, with a medium-high income level, in which 90% of the residential centres are privately owned. Care for the elderly in Spain is decentralised and each region manages the residences. In the study population, 90% of the places are privately managed and 657 professionals work in these 12 residences. Since the COVID pandemic, the clinical management of these centres has been shared by the Health Surveillance department and primary care of the National Health Public service. The health provider Fundació Assitencial Mútua de Terrassa covers the primary, secondary, and tertiary care of the people in the residences studied.

Eligibility criteria: residing in one of the residences of study of the city. Selection criteria: having a positive PCR result for SARS-CoV-2 during the study period. There were no losses.

### 2.2. Measures and Data Collection

The following information was collected from all the individuals included in the study: demographic data (age, sex, size of residence according to the number of places available), clinical data (presence of symptoms such as fever, cough or dyspnoea, basal indicators of frailty such as dependence according to the Barthel index, global deterioration scale (GDS) and glomerular filtration (GF) in ml/min/1.73 m^2^ according to the Chronic Kidney Disease Epidemiology Collaboration creatinine equation, chronic treatments (antihypertensive drugs, especially the angiotensin-converting enzyme (ACE) and angiotensin-II receptor blocker (ARB-II); antipsychotics; antidepressants; anticholinergics; GABAergics; proton pump inhibitors; nonsteroidal anti-inflammatories; benzodiazepines; oral antidiabetic; statins; inhalers; levodopa; levothyroxine; opioids), acute treatment (corticoids; hydroxychloroquine), and disease evolution (infection by SARS-CoV-2 and mortality).

Infection by SARS-CoV-2 was considered in residents providing a nasopharyngeal swab positive for SARS-CoV-2 with the use of the commercial reverse transcription PCR (AllplexTM 2019-nCoV Assay (Seegene)) which detects the viral genes E, N, and RdRp. Diagnostic tests were performed in most symptomatic and asymptomatic residents. All the staff with symptoms suspicious of COVID-19 and those asymptomatic in the contact study underwent a PCR test.

Data was retrospectively collected daily from 15 March to 11 May 2020 by eight clinicians who attended the residences. Data was collected according to guidelines of Good Clinical Practice. Review of the electronic clinical histories was made using HP-HCIS^®^, and the data were stored using Research Electronic Data Capture^®^ database software [39].

Data on the medications consumed were obtained from the Datamart app of prescription billing in pharmacies of the Catalan Health Care Services, and Anatomical Therapeutic Chemical (ATC) groups were considered.

### 2.3. Statistical Analysis

Categorical data were expressed using frequencies and percentages, while continuous data were described using mean and standard deviation (SD). A chi-square test was used to study the association between the categorical variables and the two outcomes; incidence and mortality caused by SARS-CoV-2 infection and a Student’s t-test in the case of continuous variables. The associations were also evaluated in terms of odds ratios (OR). Clinically meaningful variables showing a significant level in the univariate analysis (*p* < 0.05) were thereafter included in the multivariable logistic model. A backward stepwise method was used to identify independent risk predictors with *p* < 0.05 for the inclusion or deletion criterion. To account for within-cluster correlation and clusters defined by residences, the estimation procedure was the generalised estimating equations and the correlation structure assumed was the exchangeable. The analyses were performed using R software version 4.0.3, specifically with the geepack package (R project for statistical computing; Vienna, Austria).

## 3. Results

A total of 842 residents were included in the study and 784 who had undergone a molecular test for the diagnosis of SARS-CoV-2 were included in the analysis. Of these, 74.2% (582) were women with a mean age of 87.1 years (SD 8.10%) (Table 1).

### 3.1. Morbidity (Infection by SARS-CoV-2)

During the study period 87 individuals died, representing a mortality rate of 11.1%. Forty-four percent had a positive PCR test. Some symptom compatible with COVID-19 was reported in 33.4% of the residents, and of these, 80.9% were PCR-positive for SARS-CoV-2. Among the asymptomatic individuals, 25.5% were positive for SARS-CoV-2.

Bivariate analysis of the factors associated with infection by SARS-CoV-2 showed no association with the age and sex of the residents. On the other hand, residing in a nursing home with more than 50 places was associated with a greater risk of SARS-CoV-2 infection (nursing home of 50–99 places: OR 3.32 [1.68; 6.63]; nursing home ≥ 100 places: OR 4.01 [2.52; 6.65], *p* > 0.001) (Table 2). In addition, the greater the rate of infected staff, the higher the percentage of residents infected (Spearman coefficient = 0.85, *p* = 0.0038).

The remaining factors studied, indicators of frailty (Barthel index, GDS, GF), need for chronic treatments (antihypertensives, ACE and ARB-II; antipsychotics; antidepressants; anticholinergics; GABAergics; proton pump inhibitors; nonsteroidal anti-inflammatories; benzodiazepines; oral antidiabetics; statins; inhalers; levodopa; levothyroxine; opioids) (Appendix A) and acute treatments, were not related to infection by SARS-CoV-2. The rate of infection among staff was not related to nursing home size.

In the multivariate analysis, including the variables of age, sex, Barthel index, GF, GDS, size of nursing home, and staff infection rate, infection by SARS-CoV-2 among residents was only associated with the rate of staff infected in the nursing homes (OR = 1.07, 95% confidence interval [CI]; 1.03; *p* < 0.001).

### 3.2. Mortality by SARS-CoV-2

In relation to mortality, of the 784 residents analysed, 87 (11.1% 95%CI 8.8–13.4) died by SARS-CoV-2 during the study period.

The median time of evolution to death of the residents infected by SARS-CoV-2 was 9 days (interquartile range 5 to 14 days).

The factors related to mortality by COVID-19 in the bivariate analysis were associated with male sex, the Barthel index, and the GDS. With respect to the indexes of dependence, the percentage of deaths was greater with moderate (29.5%), severe (23%), or total dementia (40.9%). The ORs of mortality in the residents with moderate, severe, or total dependence were 3.10, 95% CI [1.44; 6.72], 2.18, 95% CI [0.98; 4.81], and 5.03, 95% CI [2.50; 10.4], respectively.

The GDS was also associated with mortality, showing that 34.5% of the 87 residents with moderate or severe dementia died (OR = 2.64, 95%CI: 1.20; 6.24).

The use of different groups of chronic medication or treatment with acute systemic corticoids and GF in patients with SARS-CoV-2 infection with a torpid evolution was not associated with mortality by COVID-19 (Appendix A). Neither were the rate of infected staff (Spearman coefficient = 0.33, *p* = 0.4) or the size of the nursing home related to mortality by COVID-19 (*p* = 0.597) (Table 3).

Multivariate analysis including the variables of age, sex, Barthel index, GF, GDS, and size of nursing home showed mortality to be related to male sex (OR = 3.37 95%CI: 2.41; 4.71) and a greater grade of independence according to the Barthel index (moderate; OR = 4.06, 95%CI; 2.79; 5.89, severe OR = 2.62, 95%CI: 2.05–3.34, and total OR = 6.10 95%CI:5.18–7.18) (Table 4).

## 4. Discussion

The COVID-19 pandemic has had a great morbidity and mortality especially in the elderly. In addition, the SARS-CoV-2 has had a great negative impact on nursing homes, with massive outbreaks being reported in care facilities all over the world [40]. Individuals living in nursing homes for the elderly present higher indexes of frailty than the general population, making them more vulnerable to health problems such as infection by SARS-CoV-2 [37,41,42,43,44,45]. It is also known that people with greater frailty have a higher mortality by COVID-19 [46]. However, there are few studies on the factors associated with infection and mortality by SARS-CoV-2 in long-term care facilities [41]. To our knowledge, this is one of the few studies to establish an association between the risk of infection by SARS-CoV-2 in residents of nursing homes for the elderly and the rate of infection of the staff of these centres. It is also one of the few studies relating mortality by SARS-CoV-2 to male sex and frailty, specifically within the residential setting of older people.

In the first wave, entry to the residencies was almost totally limited to people who were not the staff, and therefore the only possible pathway of entry of the infection was the workers themselves. Some studies have linked the risk of infection to the size of the residences [47]. Our study shows that the risk of SARS-CoV-2 infection is related to the rate of infection in workers and not to the size of the residence. Since 11 March 2020 and continuously thereafter, the health authorities implemented protocols and training activities concerning case management, isolation, the study of contacts, isolation by cohorts and sectorisation in residences, and *the* use of individual protection equipment [48]. As recommended by different organisms and international societies, it is essential to implement measures of control of transmission of SARS-CoV-2 infection, which are specifically aimed at protecting residents from staff members by early diagnosis, correct use of personal protection equipment, and sectorisation according to the residence.

In relation to mortality by SARS-CoV-2, this is one of the first studies to relate mortality to male sex and frailty measured by the Barthel index in persons residing in nursing homes [16,37], being similar to that in hospitalised patients [17,24,49,50]. Many studies have noted that ACE2 activity is higher in males than in females [51], and it has been found that higher ACE2 expression and activity in males than females may contribute to sex differences in COVID-19 infection and fatality. The relationship between mortality due to SARS-COV-2 and the Barthel index can be explained in that people who have greater functional dependence have worse health status and more comorbidities, and thus, the presence of infection is associated with a worse evolution. Another group provides the explanation that because these individuals generally require extensive assistance with activities of daily living, they are usually in close proximity with many staff members. This group explains that these patients may have higher viral load at the time of infection, and this has been found to correlate with the risk of mortality [43]. However, a worse Barthel index would also be associated with an increased risk of SARS-CoV-2 infection, and this was not observed in our cohort.

Concerning the risk of infection, there is no association with gender and age, which is consistent with what is known [47]. The grade of GF was not associated with a greater risk of infection by SARS-CoV-2 or to a greater mortality in contrast to what has been described in hospital studies such as that by the COVID-19@Spain Study Group [24]. Neither were chronic treatments with ACE or ARB-II [52] or acute treatments with corticoids [53] related to mortality, although this subgroup of patients was very small. In this sense, of note is the importance of the information provided by studies of prognostic factors of morbidity and mortality by COVID-19, by populational and community bases, in contrast to studies in the hospital setting [49]. The former can estimate the risk with a lower risk of selection bias, and this has been produced in some studies showing discordance between the factors associated and not associated with morbidity and mortality by COVID-19 [24,54].

One of the limitations of this study was its observational nature in which we could only study the association of factors with infection and mortality from SARS-CoV-2. Although the clinical evaluation of most patients included a PCR test, it was not performed in all the patients. Another limitation was that differences in the quality of infection control of various institutions were not evaluated.

The change in the populational pyramid in Western societies, with an elevated percentage of the population being above the age of 65 years and with a longer life expectancy, has led this longevity and the elevated comorbidity to produce a situation of frailty, which is especially prevalent in nursing homes for the elderly [55,56]. Likewise, these centres have been largely designed and conceived as a formal residence, mainly during the last years of life. These homes are not designed to respond to a situation of a very transmissible and highly lethal infectious epidemic. This scenario highlights the fragility of a segment of the population that is institutionalised and frequently requires complex health care when they become ill.

## 5. Conclusions

The COVID-19 pandemic has questioned the management of dependency care in nursing homes, and measures aimed at reducing the risk of infections will have to be considered to improve care for dependents. The risk of SARS-CoV-2 infection in persons living in nursing homes was related to the rate of infection among the staff of these homes, and the risk of mortality by SARS-CoV-2 was related to sex and the grade of dependence according to the Barthel index. These data may be very relevant for developing future measures aimed at the prevention of morbidity and mortality by SARS-CoV-2 or similar epidemics. These results also highlight the frailty of this population and the high mortality caused by SARS-CoV-2 infection.

## Figures and Tables

**Table 1 ijerph-18-10221-t001:** Descriptive data.

	[ALL]
	*N* = 784
SEX:	
Female	582 (74.2%)
Male	202 (25.8%)
AGE	87.1 (SD 8.10)
EXITUS:	
No death	697 (88.9%)
Exitus	87 (11.1%)
PCR SARS-CoV-2:	
Negative	439 (56.0%)
Positive	345 (44.0%)
SYMPTOMATIC:	
No	522 (64.3%)
Yes	262 (35.7%)
NURSING HOMES ACCORDING TO PLACES AVAILABLE	
<49 places	120 (15.3%)
50–99 places	61 (7.78%)
≥100 places	603 (76.9%)

**Table 2 ijerph-18-10221-t002:** Factors related to SARS-CoV-2 infection.

	[ALL]	Negative	Positive	OR [95% CI]
	*N* = 784	*N* = 439	*N* = 345	
Sex				
Female	582 (74.2%)	322 (55.3%)	260 (44.7%)	
Male	202 (25.8%)	117 (57.9%)	85 (42.1%)	0.90 [0.65; 1.24]
Age (years)	87.1 (8.10)	87.0 (8.11)	87.4 (8.11)	1.01 [0.99; 1.02]
Barthel index				
Mild or independent	319 (42.9%)	177 (55.5%)	142 (44.5%)	Ref.
Moderate	137 (18.4%)	80 (58.4%)	57 (41.6%)	0.89 [0.59; 1.33]
Severe	143 (19.2%)	82 (57.3%)	61 (42.7%)	0.93 [0.62; 1.38]
Total	145 (19.5%)	79 (54.5%)	66 (45.5%)	1.04 [0.70; 1.55]
Glomerular filtration ^1^				
≥60	422 (62.7%)	238 (56.4%)	184 (43.6%)	Ref.
59–30	219 (32.5%)	114 (52.1%)	105 (47.9%)	1.19 [0.86; 1.65]
<29	32 (4.75%)	16 (50.0%)	16 (50.0%)	1.29 [0.62; 2.69]
GDS ^2^				
Normal or subjective deterioration	161 (32.1%)	100 (62.1%)	61 (37.9%)	Ref.
Mild or moderate dementia	155 (30.9%)	88 (56.8%)	67 (43.2%)	1.25 [0.79; 1.96]
Moderate-severe or severe dementia	186 (37.1%)	99 (53.2%)	87 (46.8%)	1.44 [0.94; 2.22]
Size of nursing home				
<49 places	120 (15.3%)	97 (80.8%)	23 (19.2%)	Ref.
50–99 places	61 (7.78%)	34 (55.7%)	27 (44.3%)	3.32 *** [1.68; 6.63]
≥100 places	603 (76.9%)	308 (51.1%)	295 (48.9%)	4.01 *** [2.52; 6.65]

^1^ Glomerular filtration measured by ml/min/1.73 m^2^; ^2^ GDS: Global Deterioration Scale. ***: *p*-value < 0.001.

**Table 3 ijerph-18-10221-t003:** Factors related to mortality by SARS-CoV-2.

	[ALL]	No Exitus	Exitus	OR [95% CI]	*p*-Value
	*N* = 345	*N* = 258	*N* = 87		
Sex					0.001
Female	260 (75.4%)	207 (79.6%)	53 (20.4%)	Ref.	
Male	85 (24.6%)	51 (60.0%)	34 (40.0%)	2.60 [1.52; 4.41]	
Age (years)	87.4 (8.11)	87.3 (8.29)	87.5 (7.57)	1.00 [0.97; 1.03]	0.899
Barthel index					<0.001
Mild or independent	142 (43.6%)	125 (88.0%)	17 (12.0%)	Ref.	
Moderate	57 (17.5%)	40 (70.2%)	17 (29.8%)	3.10 [1.44; 6.72]	
Severe	61 (18.7%)	47 (77.0%)	14 (23.0%)	2.18 [0.98; 4.81]	
Total	66 (20.2%)	39 (59.1%)	27 (40.9%)	5.03 [2.50; 10.4]	
(Barthel index total ^1^)	50.0 [25.0; 75.0]	55.0 [25.0; 75.0]	35.0 [7.50; 50.0]	0.98 [0.97; 0.99]	<0.001
GF ^1^ (total)	67.0 [51.0; 80.0]	68.0 [51.0; 80.0]	64.0 [48.0; 79.0]	1.00 [0.98; 1.01]	0.518
GF ^1,2^ categories:					0.112
≥60	184 (60.3%)	144 (78.3%)	40 (21.7%)	Ref.	
59–30	105 (34.4%)	71 (67.6%)	34 (32.4%)	1.72 [1.00; 2.96]	
<29	16 (5.25%)	13 (81.2%)	3 (18.8%)	0.86 [0.18; 2.88]	
GDS ^3^					0.003
Normal or subjective deterioration	61 (28.4%)	51 (83.6%)	10 (16.4%)	Ref.	
Mild or moderate dementia	67 (31.2%)	58 (86.6%)	9 (13.4%)	0.79 [0.29; 2.15]	
Moderate-severe or severe dementia	87 (40.5%)	57 (65.5%)	30 (34.5%)	2.64 [1.20; 6.24]	
SIZE OF NURSING HOME					0.597
25–49 places	23 (6.67%)	19 (82.6%)	4 (17.4%)	Ref.	
50–99 places	27 (7.83%)	19 (70.4%)	8 (29.6%)	1.94 [0.51; 8.66]	
≥100 places	295 (85.5%)	220 (74.6%)	75 (25.4%)	1.57 [0.56; 5.70]	

^1^ Median quartile quintile 25 and 75%; ^2^ GF: glomerular filtration measured by ml/min/1.73 m^2^; ^3^ GDS: Global Deterioration Scale.

**Table 4 ijerph-18-10221-t004:** Multivariate model of factors related to mortality by SARS-CoV-2: Estimation.

	Estimation	Standard Error	Z Value	*p*-Value	OR	Lower OR	Upper OR
Sex							
Female					Ref.		
Male	1.22	0.17	50.7	<0.001	3.37	2.41	4.71
Barthel Index							
Mild or independent					Ref.		
Moderate	1.40	0.02	54.1	<0.001	4.06	2.79	5.89
Severe	0.96	0.12	60.1	<0.001	2.62	2.05	3.34
Total	1.81	0.08	468.6	<0.001	6.10	5.18	7.18

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
