# Peer review of "Risk Factors for COVID-19 Morbidity and Mortality in Institutionalised Elderly People"

_ijerph, 2021, doi:10.3390/ijerph181910221_

Round 1

Reviewer 1 Report

I want to congratulate the authors on the subject of the manuscript. In my opinion, I found the study very interesting and I think the topic is very 
necessary.
The manuscript is written in an understandable way and contains in each section the most relevant aspects of the research.
However, the manuscript must perform some important revisions. Document attachment.

Author Response

Reviewer comment

Thank you for the opportunity to review this manuscript submission. I want to congratulate the authors on the subject of the manuscript. In my opinion, I found the study very interesting and I think the topic is very necessary. The manuscript is written in an understandable way and contains in each section the most relevant aspects of the research. However, the manuscript must perform some important revisions.

RESPONSE: We thank the reviewer for the corrections and suggestions made. We have taken them into account and they have clearly helped to improve the quality of our article. Find find the point-by-point responses below.

Abstract: Little antecedents, excessive use of abbreviations. The type of study is not clear (observational, cohorts, prospective, ...).

RESPONSE: We would have liked to explain more background, however the word limit of only 250 words made this impossible. We have added a sentence. With regard to the abbreviations, we use only 5 (SARS-CoV-2, PCR, COVID-19, OR and CI). We have clarified that this was a cohort study We have removed the proportions from the results and rewritten the conclusions.

The title: Is too long (more than 15 words, little concise and overexplaining. The descriptors (keywords) of the study are not clearly identified.

RESPONSE: The new title is shorter, clearer and more concise: “RISK FACTORS FOR COVID 19 MORBIDITY AND MORTALITY IN INSTITUTIONALIZED ELDERLY PEOPLE “ The new Keywords according to Mesh terms are: Frailty, Frail Elderly, Comorbidity, COVID-19, Nursing homes, Long-Term Care.

1.Introduction:

Lack of references in the text, justifying the data and highlighting the international scope [line 59-69]

RESPONSE: We have introduced new international references and enriched this part of the introduction. “ Older people have been the most severely affected by COVID-19 in most countries [7], and those with comorbidities are at particular risk of having severe infection [8] and are at higher risk of dying as a result of the disease [9]–[11]. In one of the largest case series published so far, of 72,314 cases reported by the Chinese Centre for Disease Control and Prevention, case fatality was 8.0% in patients 70-79 years old and 14.8% in patients aged ≥80 years. Older adults appear to be more susceptible to the virus with 75% of known infections being in persons aged 50 and over. Hypertension followed by diabetes and cardiovascular diseases were the most common comorbidity seen in COVID-19 positive patients across major epicentres worldwide [12]. Overall the risk of severe COVID-19 disease or death is higher in older patients and those with previous medical conditions including COPD and cardiovascular disease as some of the most relevant predictors [13]. Males, over 65 years of age, and smokers may have a greater risk of developing critical or mortal conditions, and comorbidities such as hypertension, diabetes, cardiovascular disease, and respiratory diseases can also greatly affect the prognosis of COVID-19. [14].”

The following references have been added:

[B. Thakur et al., ‘A systematic review and meta-analysis of geographic differences in comorbidities and associated severity and mortality among individuals with COVID-19’, Sci Rep, vol. 11, no. 1, Art. no. 1, Apr. 2021, doi: 10.1038/s41598-021-88130-w.

W. Liu et al., ‘Analysis of factors associated with disease outcomes in hospitalized patients with 2019 novel coronavirus disease’, Chin Med J (Engl), vol. 133, no. 9, pp. 1032–1038, May 2020, doi: 10.1097/CM9.0000000000000775.

L. Gao et al., ‘Prognostic value of NT-proBNP in patients with severe COVID-19’, Mar. 2020. doi: 10.1101/2020.03.07.20031575.

F. Zhou et al., ‘Clinical course and risk factors for mortality of adult inpatients with COVID-19 in Wuhan, China: a retrospective cohort study’, Lancet, vol. 395, no. 10229, pp. 1054–1062, Mar. 2020, doi: 10.1016/S0140-6736(20)30566-3.

Y. Cheng et al., ‘Kidney disease is associated with in-hospital death of patients with COVID-19’, Kidney International, vol. 97, no. 5, pp. 829–838, May 2020, doi: 10.1016/j.kint.2020.03.005.

K. T. Bajgain, S. Badal, B. B. Bajgain, and M. J. Santana, ‘Prevalence of comorbidities among individuals with COVID-19: A rapid review of current literature’, Am J Infect Control, vol. 49, no. 2, Art. no. 2, Feb. 2021, doi: 10.1016/j.ajic.2020.06.213.

A. Izcovich et al., ‘Prognostic factors for severity and mortality in patients infected with COVID-19: A systematic review’, PLoS One, vol. 15, no. 11, Art. no. 11, Nov. 2020, doi: 10.1371/journal.pone.0241955.

Z. Zheng et al., ‘Risk factors of critical & mortal COVID-19 cases: A systematic literature review and meta-analysis’, J Infect, vol. 81, no. 2, Art. no. 2, Aug. 2020, doi: 10.1016/j.jinf.2020.04.021.

2. Materials and Methods:

-Better define the study design (observational? Retrospective cohort?, .. and type of sampling criteria for inclusion, exclusion, losses, ...

RESPONSE: Study design: Retrospective cohort study- Eligibility criteria: residing in one of the residences of study in the city. Selection criteria: having a positive PCR result for SARS-CoV-2 within the study period. There were no losses.

2.2 Measures and data collection: Ethical considerations: the materials need to include the code of proof of review by any ethical council or committee of the institution to use the clinical history and confidential data, authorization for the use of questionnaires, informed consent of the patients and the information sheet of the patients. Confidentiality and anonymity is not mentioned in the collection, analysis and custody of the data. The variables are not clear enough.

RESPONSE: In the section on the Institutional Review Board Statement we specify: The study was approved by the Ethical Committee of Investigation with Medication of the Hospital Mútua Terrassa on June 17, 2020, Minutes 9/2020. And in the Informed consent statement section we specify: Patient consent was waived due to the retrospective cohort study design approved by the referral ethics committee". We clarify that we followed Good Clinical Practice procedures, and thus, all data was collected, analysed and stored anonymously respecting confidentiality at all times.

It remains to define the type of study, size of the sample calculation,..

RESPONSE: There was no calculation of the sample size, the sample included all the residents of the residences of Sant Cugat del Vallès.

Data collection: describe with more precision the instruments, validity, procedures… It is not clear. Specify the procedure, it seems that the data was collected before passing the ethics committee.

RESPONSE: We have now specified that the data were retrospectively collected from March 15 to May 11, 2020.

2.3 Statistical Analysis

It is not specified how the database was, ..

RESPONSE: We have now specified that the R library geepack package was used.

3. Results

Lack of order in the presentation and design of the results, absence of figures and excess of tables with different designs. Do not repeat table data in the text [line 155].

RESPONSE: We have modified the text and tables for better understanding and to avoid repetition of the data in the text and in the tables.

4. Discussion

-There is a lack of order in the presentation of the discussion.

RESPONSE: The Discussion has been rewritten.

-This statement is not mentioned in the study objective (differences by sex and comparison in the sample, homogeneity) appears directly in the discussion [line 212].

RESPONSE: The objective of the study was to evaluate the factors related to infection and mortality and one of the individual variables studied was sex, which appears implicitly in the objective and is comment on in the Discussion. In the multivariate model, sex was not found to be significant. We also evaluated the interactions of each variable with sex and a study disaggregated by sex and it was not found to be significant either following these analyses. Therefore, we did not include it because it did not seem to provide any added value.

-The protocol of action of the workers in front of the residents is not specified (training, individual protection equipment, isolation by cohort, suspicion, type of rooms, common dining areas,… [line 223].

RESPONSE: In this respect we have added the following: “Since March 11, 2020 and continuously thereafter, the health authorities have implemented protocols and training activities concerning case management, patient isolation, the study of contacts, isolation by cohorts and sectorization in residences, and the use of individual protection equipment. “

Ref: Departament de Salut. Maneig a les residències de casos en investigació, probables o confirmats de la COVID-19. Scientia [Internet]. 2020 Mar 13 [cited 2021 Aug 17]; Available at: https://scientiasalut.gencat.cat/handle/11351/4745

-No reference is made to the limitations of the study, type of residences, health personnel, residents, services provided, type of management, material and human resources.

RESPONSE: We have added a paragraph on the limitations of the study: “One of the limitations of this study was its observational nature in which we could only study the association of factors with infection and mortality from SARS-CoV-2. Although the clinical evaluation of most patients included a PCR test, it was not performed in all the patients. Another limitation was that differences in the quality of infection control of various institutions were not evaluated.”

References: International studies are lacking.

RESPONSE: We have added some new international references and have removed some very local references.

Reviewer 2 Report

This is an interesting study on the main risk factors for COVID 19 morbidity and mortality in institutionalized elderly people. As interesting as it is and as important as its results may be for the current pandemic as well as for future situations, a major revision and restructuring of the manuscript is needed.

The methodology section, mainly the part on the selection of participants and the geolocation of the study should be properly explained. The results should all be presented disaggregated by sex so that the discussion can make a good analysis of the gender inequalities that are already evident in this pandemic. The discussion needs to be rewritten with a proper analysis and interpretation of the main results, which is currently lacking. Finally, I believe that a new literature review is needed and new citations that support the results and discussion should be included.

A revision of the English language would be desirable.

More details on the review can be found in the attached document. 

Author Response

REVIEWER 2:

This is an interesting study on the main risk factors for COVID 19 morbidity and mortality in institutionalized elderly people. As interesting as it is and as important as its results may be for the current pandemic as well as for future situations, a major revision and restructuring of the manuscript is needed.

The methodology section, mainly the part on the selection of participants and the geolocation of the study should be properly explained. The results should all be presented disaggregated by sex so that the discussion can make a good analysis of the gender inequalities that are already evident in this pandemic. The discussion needs to be rewritten with a proper analysis and interpretation of the main results, which is currently lacking. Finally, I believe that a new literature review is needed and new citations that support the results and discussion should be included.

A revision of the English language would be desirable.

RESPONSE: We thank the reviewer for the corrections and suggestions made. We have taken then into account and they have clearly helped improve the quality of our manuscript.

Please find below the point.by-point responses to your comments.

First we would like to comment that the text has been revised by a native English language speaker.

Abstract:

The abstract should be improved, especially the introductory or contextual part, and above all the conclusion, which does not provide more than a summary of the results and I think it should be concluded with some other reflection, if the authors revise the results and conclusion section, the same data are provided.

RESPONSE: We have improved the abstract and rewritten the conclusions: SARS-Cov-2 infection in institutionalised people is associated with the infection rate in nursing home workers and mortality by SARS-Cov-2 with sex and greater dependence according to the Barthel index. The control of the staff and the special attention in the measures of control of the infection especially to the people that have greater dependency have to be the keys for the future pandemic situations. Adequate management of nursing home staff and special attention to measures of infection control, especially of individuals with greater dependency, are keys for successful management of future pandemic situations.

  1. Introduction

The introduction is adequate, it explains well the problem to be addressed, there has been an adequate bibliographical review of the state of the question, but there are details that could be improved, such as those mentioned below:

  • Lines 59-79 need further bibliographic support.

RESPONSE: We have introduced new international references and enriched this part of the introduction

“Older people have been the most severely affected by COVID-19 in most countries [7], and those with comorbidities are at particular risk of having severe infection [8] and are at higher risk of dying as a result of the disease [9]–[11]. In one of the largest case series published so far, of 72,314 cases reported by the Chinese Centre for Disease Control and Prevention, case fatality was 8.0% in patients 70-79 years old and 14.8% in patients aged ≥80 years. Older adults appear to be more susceptible to the virus, with 75% of known infections being in persons aged 50 and over. Hypertension followed by diabetes and cardiovascular diseases were the most common comorbidity seen in COVID-19 positive patients across major epicentres worldwide [12]. Overall the risk of severe COVID-19 disease or death is higher in older patients and those with previous medical conditions including chronic obstructive pulmonary disease and cardiovascular disease as some of the most relevant predictors [13]. Males, age greater than 65 years, and smokers may present a greater risk of developing critical or mortal conditions, and comorbidities such as hypertension, diabetes, and cardiovascular disease, and respiratory diseases can also greatly affect the prognosis of COVID-19. [14]. “

The following references have been added:

[B. Thakur et al., ‘A systematic review and meta-analysis of geographic differences in comorbidities and associated severity and mortality among individuals with COVID-19’, Sci Rep, vol. 11, no. 1, Art. no. 1, Apr. 2021, doi: 10.1038/s41598-021-88130-w.

  1. Liu et al., ‘Analysis of factors associated with disease outcomes in hospitalized patients with 2019 novel coronavirus disease’, Chin Med J (Engl), vol. 133, no. 9, pp. 1032–1038, May 2020, doi: 10.1097/CM9.0000000000000775.
  2. Gao et al., ‘Prognostic value of NT-proBNP in patients with severe COVID-19’, Mar. 2020. doi: 10.1101/2020.03.07.20031575.
  3. Zhou et al., ‘Clinical course and risk factors for mortality of adult inpatients with COVID-19 in Wuhan, China: a retrospective cohort study’, Lancet, vol. 395, no. 10229, pp. 1054–1062, Mar. 2020, doi: 10.1016/S0140-6736(20)30566-3.
  4. Cheng et al., ‘Kidney disease is associated with in-hospital death of patients with COVID-19’, Kidney International, vol. 97, no. 5, pp. 829–838, May 2020, doi: 10.1016/j.kint.2020.03.005.
  5. T. Bajgain, S. Badal, B. B. Bajgain, and M. J. Santana, ‘Prevalence of comorbidities among individuals with COVID-19: A rapid review of current literature’, Am J Infect Control, vol. 49, no. 2, Art. no. 2, Feb. 2021, doi: 10.1016/j.ajic.2020.06.213.
  6. Izcovich et al., ‘Prognostic factors for severity and mortality in patients infected with COVID-19: A systematic review’, PLoS One, vol. 15, no. 11, Art. no. 11, Nov. 2020, doi: 10.1371/journal.pone.0241955.
  7. Zheng et al., ‘Risk factors of critical & mortal COVID-19 cases: A systematic literature review and meta-analysis’, J Infect, vol. 81, no. 2, Art. no. 2, Aug. 2020, doi: 10.1016/j.jinf.2020.04.021.

  • Lines 104-106. I don't quite understand the introduction of an American study in the introduction because sociologically in Spain the elderly are not cared for in the same way as in America. The pension system is not the same either and neither are the socio-demographic characteristics, I think this paragraph is not relevant and should be excluded from the introduction. In any case, it would be more appropriate to include a study of another country with similar characteristics to Spain.

RESPONSE: We have removed the paragraph about the North American study and have added a Spanish study:

“A Spanish study on the characteristics of nursing homes found that the model of nursing homes best prepared to manage COVID-19 was that of public nursing homes with less than 25 places.”

Ref: E. Barrera-Algarín, F. Estepa-Maestre, J. L. Sarasola-Sánchez-Serrano, and J. C. Malagón-Siria, ‘[COVID-19 and elderly people in nursing homes: Impact according to the modality of residence]’, Rev Esp Geriatr Gerontol, vol. 56, no. 4, pp. 208–217, Aug. 2021, doi: 10.1016/j.regg.2021.02.003))

  1. Materials and Methods

2.1 Study population

This section needs to be substantially improved; only Spanish readers are aware of the location of the study population. It is necessary to explain the geolocation as well as the main characteristics of the study population and how elderly care is organized in this Autonomous Community. In Spain there is a decentralization of elderly care as well as health care. I think that this section should explain how this care is organized and whether the residence is privately or publicly managed, the number of staff, the number of beds, the number of dependent and independent residents, etc.

RESPONSE: We have improved this section with a better contextualization as follows: 

“The study population included all the residents (842 persons) of the 12 nursing homes for the elderly in the municipality of Sant Cugat del Vallès, a city near Barcelona, in the autonomous region of Catalonia, Spain. The population is of about 90,000 inhabitants, with a medium-high income level, and 90% of the residential centres are privately owned. Care for the elderly in Spain is decentralized and each region manages the residences. In the study population, 90% of the places are privately managed and 657 professionals work in these 12 residences. Since the COVID pandemic the clinical management of this centres have been shared by the Health Surveillance Department and primary care of the Public National Health service. The health provider Mútua de Terrassa covers the primary, secondary and tertiary care of the patients in the residences studied.”

2.2 Measures and data collection

Line 118. Spelling errors, please proofread the entire document to avoid such errors.

RESPONSE: Corrected.

Something that is not covered in this section is who collected the data, and were the data collectors trained to avoid variability?

RESPONSE: We have stated that the data were collected daily by 8 clinicians who attended the residence.

Another set of questions that arises when reading this section is how were the participants and residences recruited?

RESPONSE: The retrospective cohort design allowed all the institutionalised elderly people to be included in the study. We have added: Eligibility criteria: residing in one of the residences of study of the city. Selection criteria: having a positive PCR study result for SARS-CoV-2 during the study period. There were no losses.

Was informed consent sought and was the study evaluated by an ethics committee?

RESPONSE: In the section on the Institutional Review Board Statement we specify: “The study was approved by the Ethical Committee of Investigation with Medication of the Hospital Mútua Terrassa on June 17, 2020, Minutes 9/2020”. And in the Informed consent statement section we specify: “Patient consent was waived due to the retrospective cohort study design, approved by the referral ethics committee".

According to the retrospective nature of the study, the Ethics committee waived the need for informed consent.

We clarify that we followed Good Clinical Practice procedures and therefore all the data was collected, analysed and stored anonymously respecting confidentiality at all times.

Line 118. Presence of symptoms. Clarify which symptoms were taken into account

RESPONSE: We have added: “clinical data (presence of symptoms such as fever, cough or dyspnoea,...”.  

2.3 Statistical Analysis:

Please indicate the R library that was used for the study.

RESPONSE: We have introduced de R library in material.

  1. Results

Lines 153-160. In this section we have data that are repeated in both percentages and proportions, please remove repeated data.

RESPONSE: We have removed the proportions and have kept the percentages.

In addition, the same data as in table 1 is repeated. It is not appropriate to repeat the same data in text and tables, it is redundant, please select the most important data to be reflected in the text. Readers can then go to the corresponding table.

Lines 164-170. The 95% CI are missing throughout the paragraph.

RESPONSE: The 95% CIs have been added.

Lines168-169. It is not visible in table two where the Spearman coefficients and their p. In that line express reference is made to that table. A clarification or rectification would be desirable.

RESPONSE: We have moved this line after the reference in table 2.

Lines 177-179, lines 187-191. Please note that if the data referred to are in table 2 this should be reflected in this paragraph. Also check the writing of the CI because either data is missing or is poorly written and is not understood as it is expressed.

RESPONSE: We have recovered the missing CI value and have specified the 95%CI.

In the multivariate analysis, infection by SARS-CoV-2 among residents was only associated with the rate of staff infected in the nursing homes (OR=1.07, 95% CI; 1.03; 1.11, P<0.001).

Source:

Model 2

Estimate

Std. Error

z value

P value

OR

Lower OR

Upper OR

T.INFECT.TREBALL

0.07

0.017

16.0

<0.001

1.07

1.03

1.11

Conclusion of the results: I think all the results need to be reordered and rewritten. As they are presented, they are confusing and lack more detailed interpretation.

RESPONSE: We have improved the Results section.

  1. Discussion

The discussion needs to be revised and provide further literature to justify the findings. In addition, certain specific aspects of this section are detailed below:

Lines 223-230. What is reflected in this paragraph is correct. After reading it, I have a thought: the authors of this article have not considered the number of workers in the nursing home and the number of COVID 19 infections in these workers as a risk factor for infections in residents. Can they give an explanation for this, since the paragraph presented here assumes that it has also been an important risk factor in this pandemic?

RESPONSE: We considered the number of workers in the nursing home and the number of COVID 19 infections among these workers but did not specify these data in the section on Material and Methods. They have now been added.

The explanation states that after the restriction of family visits, the staff became the only main entry point for the transmission of this viral infection to the residents.

Lines 231-235. I think you have to be careful what you discuss in that paragraph. I think you must make a proper interpretation of the size of the residence and the risk of infection. I agree that your results reflect this association, but at no point have you reflected the number of workers in the homes analyzed. I do not agree that the larger the nursing home, the higher the risk of infection if the ratio of professional staff to staff is adequate. Moreover, if there are bubble groups, there is no reason for this situation to arise, or if the professionals are given the appropriate PPE, they are trained in their use and there are enough workers. Furthermore, a comparison is made with an American study where the model of care for dependency is not comparable with our country. I think this should be taken into account and interpreted correctly.

RESPONSE: Thanks for the explanation; the reviewer is right, and we have reconstructed the argument.

“In the first more intense wave, entry to the residences was almost totally limited to people who were not staff, and therefore the only possible pathway of infection entry was the workers themselves. Some studies have linked the risk of infection to the size of the residences [46]. Our study shows that the risk of SARS-CoV-2 infection is related to the rate of worker infection and not to the size of the residence. Since March 11, 2020 and continuously thereafter, the health authorities have implemented protocols and training activities concerning case management, isolation, the study of contacts, isolation by cohorts and sectorization in residences, and the use of individual protection equipment [47]. As recommended by different organisms and international societies, it is essential to implement measures of control of transmission of SARS-CoV-2 infection, which are specifically aimed at protecting residents from infection by staff members by early diagnosis, correct use of personal protection equipment and sectorization according to the residence [6], [26].”

In relation with lines 164-165 of results, where the authors said “Bivariate analysis of the factors aasociated with…” I do not know if the authors can provide an explanation for this finding, especially since in the first wave a clear relationship between age and COVID morbidity and mortality has been seen. Likewise, the authors also say that they find no differences by sex in morbidity. Do they have any explanation in this respect? What do the disaggregated data provide?

RESPONSE: The problem with the association with age is that our sample was very old and by not being able to compare it with the rest of the younger population, age is not significant.

There was no association with gender and sex and this is consistent with what is known (Ref.Mukherjee S, Pahan K. Is COVID-19 Gender-sensitive? J Neuroimmune Pharmacol. 2021 Mar;16(1):38-47. doi: 10.1007/s11481-020-09974-z. Epub 2021 Jan 6. PMID: 33405098; PMCID: PMC7786186.).

In the explanation about the difference of the gender and mortality for COVID-19 , we have added:  “Many studies have noted that ACE2 activity is higher in males than in females, and it has been found that higher ACE2 expression and activity in males than females may contribute to sex differences in COVID-19 infection and fatality.”.  

Ref: Gargaglioni LH, Marques DA. Let's talk about sex in the context of COVID-19. J Appl Physiol (1985). 2020 Jun 1;128(6):1533-1538. doi: 10.1152/japplphysiol.00335.2020. Epub 2020 May 21. PMID: 32437244; PMCID: PMC7303729.)

The discussion section does not include the limitations of the study, it is an important part that should be included. A better literature review should be conducted as there are published studies that can support the discussion. It should also be noted that this is one of the first studies with these results.

RESPONSE: We add a paragraph about the limitations of the study:One of the limitations of this study was it observational nature in which we could only study the association of factors with infection and mortality from SARS-CoV-2. Although the clinical evaluation of most patients included a PCR test, it was not performed in all the patients. Another limitation was that differences in the quality of infection control of various institutions were not evaluated.”

  1. Conclusions

The conclusions need to be developed further. It is clear that the COVID 19 pandemic has called into question the management of dependency care in Spain, but the authors should draw more conclusions from their study, what public health measures can be put in place, what social measures are necessary to improve dependency care, what consequences this pandemic has had on social health or mental health for residents, for example. I think there are strong points that need to be addressed.

RESPONSE: We have rewritten the conclusions:

“The COVID 19 pandemic has questioned the management of dependency care in nursing homes and measures aimed at reducing the risk of infections will have to be considered to improve care for dependence. We found that the risk of SARS-CoV-2 infection in persons living in nursing homes was related to the rate of infection among the staff of these homes, and the risk of mortality by SARS-CoV-2 was related to sex and the grade of dependence according to the Barthel index. These data may be very relevant for developing future measures aimed at the prevention of morbidity and mortality by SARS-CoV-2 or similar epidemics. These results also highlight the frailty of this population and the high mortality caused by SARS-CoV-2 infection.”

Tables and Figures:

Table 1. Table 1 as it is presented does not provide much information and as noted above, its information is repeated in text at the beginning of the results. It makes NO sense to put a column with the N when it is all the same. It would be convenient to present this table disaggregated by sex. It is important that a commitment is made in the research to disaggregate the analyses by sex in order to analyze gender differences in health.

Why does this table not include a complete descriptive with all the variables that have been considered for the study? Perhaps it would be convenient to make a usual presentation of the results, first a descriptive of all the variables and then advance in complexity, and not to mix in the bivariate and multivariate descriptive analyses of the morbidity variables, for example (table 2).

All tables. Men and women do not get sick in the same way and the risk factors are not the same for both. Would it be possible for all tables to be presented disaggregated by sex? In as many tables as possible, it is desirable to present the CI. I agree on the importance of presenting the p-value, but a CI also helps us to assess statistical power through its width. If the authors want to reduce space the p-values can be presented with asterisks.

RESPONSE: In the multivariate model, sex was not found to be significant. We also evaluated the interactions of each variable with sex and a study disaggregated by sex, and it was found to not be significant either following these analyses. Therefore, we did not include it because it did not seem to provide any added value. These are the stratified analysis by sex:  

Table 2_factors related to SARS-CoV-2-infection

Summary descriptive tables

FEMALE

MALE

Negative

N=322

Positive

N=260

OR [95% CI]

Negative

N=117

Positive

N=85

OR [95% CI]

Age

87.6 (7.84)

88.3 (7.46)

1.01 [0.99;1.04]

85.3 (8.61)

84.4 (9.27)

0.99 [0.96;1.02]

Barthel Index

Mild or independent

123 (55.4%)

99 (44.6%)

Ref.

54 (55.7%)

43 (44.3%)

Ref.

Moderate

62 (55.9%)

49 (44.1%)

0.98 [0.62;1.55]

18 (69.2%)

8 (30.8%)

0.57 [0.21;1.40]

Severe

57 (53.3%)

50 (46.7%)

1.09 [0.68;1.73]

25 (69.4%)

11 (30.6%)

0.56 [0.24;1.24]

Total

64 (55.7%)

51 (44.3%)

0.99 [0.63;1.56]

15 (50.0%)

15 (50.0%)

1.25 [0.55;2.88]

Glomerular Filtration:

>=60

175 (55.4%)

141 (44.6%)

Ref.

63 (59.4%)

43 (40.6%)

Ref.

59-30

96 (54.9%)

79 (45.1%)

1.02 [0.70;1.48]

18 (40.9%)

26 (59.1%)

2.10 [1.03;4.37]

<29

12 (48.0%)

13 (52.0%)

1.34 [0.59;3.10]

4 (57.1%)

3 (42.9%)

1.11 [0.20;5.57]

Global Deterioration Scale

Normal or subjective deterioration

62 (59.0%)

43 (41.0%)

Ref.

38 (67.9%)

18 (32.1%)

Ref.

Mild or moderate dementia 

62 (54.4%)

52 (45.6%)

1.21 [0.71;2.07]

26 (63.4%)

15 (36.6%)

1.22 [0.51;2.86]

Moderate-severe or severe dementia 

83 (54.6%)

69 (45.4%)

1.20 [0.72;1.99]

16 (47.1%)

18 (52.9%)

2.35 [0.98;5.76]

Size of Nursing Home

<=49 places

59 (76.6%)

18 (23.4%)

Ref.

38 (88.4%)

5 (11.6%)

Ref.

50-99 places

27 (54.0%)

23 (46.0%)

2.76 [1.28;6.05] **

7 (63.6%)

4 (36.4%)

4.20 [0.82;21.0]

>=100 places

236 (51.9%)

219 (48.1%)

3.02 [1.76;5.43] **

72 (48.6%)

76 (51.4%)

7.75 [3.12;23.9] *

* P value<0.05

c2.all<-compareGroups(EXITUS~EDAT+Barthel.cat.rec+FG+FG_new+GDS_rec+Tipologia.Residencia_rec,byrow=TRUE,method=4,data= subset(residencia,PCR.COVID.19=="Positiva"))

Table 3: Factors related to mortality by SARS-CoV-2.

Summary descriptive tables

FEMALE

MALE

No exitus N=207

Exitus N=53

OR [95% IC]

No exitus N=51

Exitus N=34

OR [95% IC]

Age

88.1 (7.72)

89.4 (6.28)

1.03 [0.98;1.07]

84.4 (9.84)

84.5 (8.48)

1.00 [0.95;1.05]

Barthel Index

Mild or independent

93 (93.9%)

6 (6.06%)

Ref.

32 (74.4%)

11 (25.6%)

Ref.

Moderate

36 (73.5%)

13 (26.5%)

5.45 [1.97;16.9]*

4 (50.0%)

4 (50.0%)

2.84 [0.55;14.7]

Severe

42 (84.0%)

8 (16.0%)

2.91 [0.94;9.57]

5 (45.5%)

6 (54.5%)

3.38 [0.83;14.5]

Total

31 (60.8%)

20 (39.2%)

9.66 [3.72;28.9]*

8 (53.3%)

7 (46.7%)

2.50 [0.71;8.83]

Glomerular filtration total

68.0 [51.0;80.0]

64.0 [47.8;76.8]

0.99     [0.98;1.01]

65.0 [52.5;80.5]

63.0 [50.0;85.0]

1.00 [0.97;1.02]

Glomerular Filtration:

>=60

116 (82.3%)

25 (17.7%)

Ref.

28 (65.1%)

15 (34.9%)

Ref.

59-30

58 (73.4%)

21 (26.6%)

1.68 [0.86;3.26]

13 (50.0%)

13 (50.0%)

1.85 [0.68;5.11]

<29

11 (84.6%)

2 (15.4%)

0.89 [0.12;3.67]

2 (66.7%)

1 (33.3%)

0.99 [0.03;13.2]

Global Deterioration Scale

Normal or subjective deterioration

37 (86.0%)

6 (14.0%)

Ref.

14 (77.8%)

4 (22.2%)

Ref.

Mild or moderate dementia 

48 (92.3%)

4 (7.69%)

0.52 [0.12;2.03]

10 (66.7%)

5 (33.3%)

1.71 [0.35;8.94]

Moderate-severe or severe dementia 

48 (69.6%)

21 (30.4%)

2.64 [1.00;7.91]*

9 (50.0%)

9 (50.0%)

3.32 [0.80;16.1]

Size of Nursing Home

<=49 places

15 (83.3%)

3 (16.7%)

Ref.

4 (80.0%)

1 (20.0%)

Ref.

50-99 places

17 (73.9%)

6 (26.1%)

1.71 [0.37;9.85]

2 (50.0%)

2 (50.0%)

3.28 [0.17;145]

>=100 places

175 (79.9%)

44 (20.1%)

1.21 [0.37;5.64]

45 (59.2%)

31 (40.8%)

2.49 [0.32;70.2]

 * P value<0.05

Infection rates in staff members of nursing homes.

SARS-CoV-2 infection

R Spearman

P value

All

0.856

0.0004

Female

0.881

<0.001

Male

0.877

<0.001

Mortality by SARS-CoV-2

R Spearman

P value

All

0.763

0.004

Female

0.806

0.002

Male

0.622

0.003

We have corrected the 95% CI and the p-value is shown with asterisks: *: p-value <0.05; **: p-value <0.01; : p-value <0.001.

Table 2. Table 2 presents the factors associated with COVID infection, but in the explanatory paragraph it talks about bivariate and multivariate analysis. What is presented in this table? Which of the two analyses is presented? Also, what variables have been considered for the multivariate analysis?

RESPONSE: We have presented the table of the bivariate analysis. For better understanding we have included the explanation of the multivariate analysis, specifying the variables that were introduced in the mode below the table.

Again, why have these analyses not considered a sex-disaggregated analysis when much has already been said about gender inequalities in the COVID pandemic?

Table 3. The same comment as reflected for table 3.

Table 4. This table presents the multivariate analysis of the factors that have influenced COVID 19 mortality. Can we explain which variables have been considered in these analyses?

RESPONSE: We have added the explanation of the multivariate analysis, specifying the variables  introduced in the model

All tables are self-explanatory. It would be a good idea to include footnotes to the tables explaining the models used.

RESPONSE: Thanks. This was not taken into account.

  1. References

As indicated above, it is necessary to review the bibliographical citations and include others to support the arguments of this research.

Likewise, care must be taken as there are terms in Spanish when the language of the manuscript is in English. If possible, include the DOI or URL in all bibliographic citations.

RESPONSE: We have expanded the bibliography to provide greater support to the argumentation of the research.

We have corrected the citations in Spanish language and have added the DOIs.

Reviewer 3 Report

  1. This study collects data from 12 nursing homes and 842 residents to explore the factors related to morbidity and mortality by COVID-19. When the world is still threatened by COVID-19, this research is a preliminary article with value.
  2. This study used the backward stepwise method to identify independent risk predictors with P <0.05 for the inclusion or deletion criterion. This method is less empirical, and it is recommended that the selection of research variables should be based on research theory and hypothesis
  3. The subjects of this study are from 12 nursing homes. In addition to the size of the institution and the infection rate of staff, is there any heterogeneity among other institutions that may affect the results of the study? Such as the quality of infection control of various institutions, etc.
  4. Although there are few references for this research topic at present, the discussion content is still relatively weak. It is suggested that the research results can be discussed more. For example, why does the Barthel index have no dose responses on mortality by COVID-19? Why does residents with moderate or severe dementia have a higher risk of mortality by COVID-19?

Author Response

We thank the reviewer for the corrections and suggestions made, which have clearly helped to improve the quality of the manuscript.

Below please find the point-by-responses to your comments.

  1. This study collects data from 12 nursing homes and 842 residents to explore the factors related to morbidity and mortality by COVID-19. When the world is still threatened by COVID-19, this research is a preliminary article with value.
  2. This study used the backward stepwise method to identify independent risk predictors with P <0.05 for the inclusion or deletion criterion. This method is less empirical, and it is recommended that the selection of research variables should be based on research theory and hypothesis

  1. The subjects of this study are from 12 nursing homes. In addition to the size of the institution and the infection rate of staff, is there any heterogeneity among other institutions that may affect the results of the study? Such as the quality of infection control of various institutions, etc.

RESPONSE: We did not study the differences centre by centre because the unit was the resident and not the residence, although we have related the resident with infection rate in workers . The quality of infection control in each nursing home may have varied and may justify differences but we were not able to measure it.

  1. Although there are few references for this research topic at present, the discussion content is still relatively weak. It is suggested that the research results can be discussed more. For example, why does the Barthel index have no dose responses on mortality by COVID-19?

RESPONSE: It is true that the Barthel Index does not follow a gradual dose because the "severe" OR is 2.62. We have no explanation for this because the ratios are similar in each group. We were also surprised by this result considering that the number of people in each Barthel index was similar. One possible explanation is that the sample sizes were small and the CI wide.

  1. Why does residents with moderate or severe dementia have a higher risk of mortality by COVID-19?

RESPONSE: The relationship between dementia and COVID-19 mortality has been reported, we have not found studies that link the severity of dementia with mortality. While the relationship between dementia and COVID-19 mortality has been reported, we found no study linking the severity of dementia with mortality.  At the functional level and due to the usual evolution of dementias, the greater the severity, the greater the frailty of the patient and the greater the risk of worse evolution.

Round 2

Reviewer 1 Report

Thank you for the opportunity to review this manuscript submission. I want to congratulate the authors on the subject of the manuscript. In my opinion, I found the study very interesting and I think the topic is very necessary. After the reviews carried out, the manuscript is written in an understandable way and contains in each section the most relevant aspects of the research

Reviewer 2 Report

Dear Editor and authors, the revisions you have made have added to the quality of your manuscript, which can be accepted for publication.

Reviewer 3 Report

1. This article has been revised significantly, and the content is richer than the previous version.
2. English grammar still needs to be professionally edited, and it is recommended to publish after editing.